# Development and Evaluation of a Preceptor Education Program Based on the One-Minute Preceptor Model: Participatory Action Research

**DOI:** 10.3390/ijerph182111376

**Published:** 2021-10-29

**Authors:** Hye Won Jeong, Deok Ju, Myoung Lee Choi, Suhyun Kim

**Affiliations:** 1Department of Nursing, Chonnam National University Hospital, Gwangju 61469, Korea; y2k331646@gmail.com (H.W.J.); juyulianna@hanmail.net (D.J.); choimy6@naver.com (M.L.C.); 2Department of Nursing, Nambu University, Gwangju 62271, Korea

**Keywords:** education, one-minute preceptor model, preceptorship, participatory action research, nursing, education program, communication ability, clinical nurse, mixed research method

## Abstract

This participatory action research study was conducted to confirm the implementation process and effect of developing and applying a preceptor education program based on the One-Minute Preceptor Model to foster the competence of preceptor clinical nurses. The study was conducted for eight weeks from March 2020 on 30 preceptor nurses in South Korea. Nursing standards were developed for two weeks and six cycles (comprising four stages) were performed. Data collection was integrated using both quantitative and qualitative approaches. For quantitative data, the Clinical Core Competency of Preceptor (CCCP) and General Communication Competence Scale (GICC-15) results were collected from preceptor nurses through questionnaires. Reflection journals of nurses’ experiences were also analyzed through content analysis and frequency of keywords using WordClouds. There was no significant change in CCCP or GICC-15 results among preceptor nurses. However, nurses’ experiences were associated with the growth and development of competencies such as evidence-based practice, quality feedback, and self-reflection. The program was effective in developing nurse competencies. Therefore, it is necessary to encourage One-Minute Preceptor Model activities among preceptors through an action research approach and to actively support research and practice in clinical settings, as well as to provide organizational and systematic support.

## 1. Introduction

The turnover rate of new nurses in Korea rose sharply from 28.7% in 2014 to 42.7% in 2018; it increased by 45% in 2021 due to the COVID-19 pandemic [1]. New nurses find the rapid bed turnover, patient disease severity, and increase in workload challenging [2,3]. The high turnover rate of new nurses not only causes huge losses to medical institutions in terms of costs, time, and workforce consumption for hiring new personnel, but also threatens patient safety [4]. Practical alternatives to address this issue are essential.

Many hospitals provide preceptorships to train, guide, and mentor new nurses [5]. Effective preceptorships can convert new nurses into professional practitioners and improve their learning competency through clinical guidance and role modeling [6,7]. A well-trained preceptor nurse provides psychological stability to new nurses, improves job satisfaction, and promotes organizational socialization [6]. This in turn helps new nurses adapt to work [7] and lowers their turnover intention [8]. However, the role of an ideal preceptor in the clinical field is difficult [5,9,10]. Preceptor nurses inevitably experience increased workload and stress because they must teach new nurses while performing routine tasks [5,9,11]. Their workload and stress not only reduce the quality of education but also threaten patient safety due to burnout and shorter patient contact times [4,12]. Moreover, many clinical nurses feel that their preceptor nurses lack educational competence and teaching effectiveness [11]. In addition, preceptorship education requires an additional workforce to maintain and manage the curriculum, and many hospitals may not be inclined to invest in this area [9,10].

However, in Korea, there is a big difference in the operation of preceptorship education depending on the size and characteristics of hospitals [12]. Most hospitals have seen the difficulties of preceptors teaching new nurses as problems to be solved by individuals [11]; therefore, research to understand the organizational approach and their experiences has recently begun [11,12,13]. As such, the development of a systematic and standardized preceptor education program is urgently required at a time when the shortage of nurses is becoming increasingly serious [14].

To address the issues of nurse shortage, the Korean government, in 2019, implemented the education-only nurse support project [15]. The role of the nurse in charge of education is to strengthen the clinical competence of nurses and improve their workplace, thereby reducing the turnover of new nurses [15,16]. Studies on the effects of this project are currently limited because education specialist nurses such as these do not know how to educate and help other nurses [15]. Education specialist nurses are not specially trained by the government or hospitals but are selected by the recommendation of the nursing department based on their clinical experience [15,16]. However, given that this resource exists, preceptorship programs should consider collaborating with the nurse in charge of education to improve the performance of preceptor nurses.

Another resource is the One-Minute Preceptor (OMP), a preceptor training model developed for clinicians to enhance their role as educators within the busy clinical environment to train medical residents and students without interfering with their work [17,18]. The OMP model consists of five stages to promote teaching strategies and the effectiveness of education [17,18,19,20]. The first step is for the preceptor to make a mutual commitment related to education at the first meeting with the preceptee. At this stage, the preceptor should devise a way for the preceptee to effectively learn clinical practice. The second step is for the instructor to adjust the practical training by evaluating the learning process according to the preceptee’s level of understanding. The third step is to teach the preceptee general rules, such as basic skills, and to share their expertise. The last two steps are to provide feedback. The fourth step is for the preceptor to effectively provide positive feedback concerning learners’ good behavior. The fifth step is to provide constructive feedback, such as correcting learners’ mistakes and making recommendations for improvement.

As a teaching tool, the OMP model is effective not only for simple knowledge transfer, but also for teaching new things that learners want to learn [19]; that is, it not only allows learners to have an active and positive learning attitude, but also improves the efficiency and teaching skills of instructors [17,20]. The OMP model is used to improve clinical reasoning and preceptor feedback skills in the clinical field. It is preferred over traditional education methods [20]. Above all, traditional teaching methods provided only one-time education to preceptors as a lecture method before starting with their first preceptor [21].

However, previous studies report that preceptors who received such education experience difficulties in communication and limitations in their competence while teaching new nurses [5,9,11]. Communication skills are essential for preceptors who teach new nurses professional nursing knowledge and skills because they need to be able to show self-disclosure, empathy, social relaxation, assertiveness, concentration, interaction management, expressiveness, and support [10,11,12]. Preceptor education based on the OPM model not only has a positive effect on providing feedback or interacting but also reports the effectiveness of preceptor education [20,21]. The authors of this study propose that a similar OMP-based education program be designed to develop the competency of preceptor nurses. In addition, they believe that engaging education and preceptor nurses in the design would help develop a practice-based training program.

To test the viability of applying the OMP model to develop a preceptor education program (PEP), this study employed a participatory action research (PAR) method. In PAR, the researcher does not conduct research on participants; rather, the researchers solve problems and conduct the research with participants [21,22]. Participants of PAR face and recognize their problems and become reflective practitioners [22,23]. Although it is not easy for participants to engage in action research [24], the process of advancing by fundamentally criticizing their habitual work methods through reflection enables problem-solving [21,22,23]. PAR has been applied in other studies on preceptor nurses in Korea [12,13,25,26]; however, these studies have not focused on developing and measuring the effectiveness of a preceptor training program. Studies have been conducted on the experiences of nurses who participated in the preceptorship of new nurses (the preceptors [9,11]), as well as the new nurses’ (preceptees) experiences of preceptorship [27]. However, these studies have not used PAR to solve the problem of developing and applying practical programs that can foster the competencies of nursing educators and preceptor nurses.

The purpose of the study is to develop and apply a PEP based on the OMP model and to confirm the implementation process and its effects through PAR. The specific research questions of this study are as follows:
What is the implementation process for the development and application of the OMP model-based PEP through PAR?What effect does the PEP that is based on the OMP model through PAR have on preceptor nurses?What are the experiences of preceptor nurses who participated in the development and application of the OMP model-based PEP through PAR?

## 2. Materials and Methods

### 2.1. Study Design

This PAR study based on the conceptual model of Zuber–Skerritt and colleagues [28,29] involved collecting and analyzing data in an integrated research method through a mixed-method approach while going through the plan, action, evaluation, and reflection stages (Figure 1).

### 2.2. Participants

The participants were 30 preceptor nurses at tertiary hospitals. All participants were aware of the purpose of the study and voluntarily agreed to participate. All participants were informed that they could withdraw from this study at any time. The preceptor nurses were those who had at least three years of clinical practice experience and had been working in the current department for at least one year. Through the first conference, all participants built up their roles. For action research, facilitators were necessary in addition to the research participants [21]. Four researchers and 4 nurses in charge of education were facilitators of the preceptors’ activity to improve their competency. The researchers consisted of one nurse with a doctor of nursing, the head nurse of the nursing department, the head nurse of the nursing education department, and one professor at a nursing college. Nurses in charge of education were government-appointed nurses with more than five years of clinical experience and were working in the education department. The required sample size for evaluating the effect of the PEP on preceptor nurses was calculated using G*Power 3.1.9.4: significance level (α = 0.05), effect size (d = 0.5), and power (1-ß = 0.08). The minimum number of participants required was 27; however, 30 people were included to account for a 10% dropout rate. This study was conducted over a period of eight weeks from March 2020 after receiving Institutional Review Board (IRB) approval.

### 2.3. Research Procedure

#### 2.3.1. Development of the PEP

The tertiary general hospital with which the first author is affiliated selects approximately 300 new nurses every year. All new nurses are expected to complete an eight-week preceptorship program. For PAR, facilitators and 30 preceptor nurses had the first meeting in a comfortable atmosphere with icebreaking activities. Brainstorming was used to decide the ground rules of active participation. The goal of the PAR was set according to the priority results of these activities.

Facilitators and preceptor nurses met once per week for eight weeks. The purpose of the PAR team’s activities was to develop and apply a PEP to foster the competence of preceptor nurses. Prior to the development of the PEP for preceptor nurses, the education specialist nurses analyzed preceptor nurses’ evaluations of participation and identified problems and points for improvement. These aspects included lack of time, heavy workload, and lack of a new standardized training plan. To solve this problem, the PAR team members devised a theoretical framework of the PEP based on the OMP model after reviewing the literature. The four researchers and education specialist nurses on the PAR team took on the role of facilitators of the overall PAR team actively. The education specialist nurses held a conference every day for preceptor nurses. They attempted to develop standards for nursing practice with preceptor nurses. The education specialist nurses conducted a systematic literature review and developed the standards by reflecting the flow of doctors’ prescriptions and nursing work that had mainly been performed in the department through continuous modifications and supplements with preceptor nurses. Facilitators recruited content experts from the department and verified the content validity of the preceptor training materials. (A content validity index score ≥0.80 was required.) Preceptor nurses participated in the final review and revision of the standardized nursing tasks.

The developed PEP was constructed by incorporating an OMP [[17],[18],[19]，[20]]. The OMP model consisted of five stages to promote teaching strategies and the effectiveness of education. The first step was for the preceptor to make a mutual commitment related to education at the first meeting with the new nurse. At this stage, the preceptor should devise a way for new nurses to effectively learn clinical practice. In addition, a standard draft for nursing practice for new nurses was prepared. The second step was for the instructor to adjust the practical training by evaluating the learning process according to the level of understanding of the new nurse. The third step was to teach new nurses general rules, such as basic nursing skills, and to share their expertise. The last two steps were to provide feedback. The fourth step was for the preceptor to effectively provide positive feedback concerning learners’ good behavior. The fifth step was to provide constructive feedback, such as correcting learners’ mistakes and making recommendations for improvement.

#### 2.3.2. Application of PEP

After development of the standard draft for nursing practice and PEP based on the OMP model, the actual application of PEP was performed every week over six sessions. PAR facilitators supported the preceptors to guide new nurses by organizing the educational curriculum and designing educational materials through weekly conferences. The weekly conference, supported by the Ministry of Nursing, was held in the training room after the preceptors’ daily duties were complete. While reflecting on their experiences, the preceptor nurses wrote the following in their diaries: their educational goals for each week, the content of the training, what they had done well, and what they lacked. Education specialist nurses conducted a mini lecture every six weeks, and preceptor nurses reviewed the contents they already knew and checked the educational goals of the week. The preceptor nurses applied the results of the discussion and contents of learning made in the PEP based on the OMP model to the new nurses, and they shared their weekly experiences with the PAR facilitators to find strategies to solve problems.

### 2.4. Data Collection and Analysis

#### 2.4.1. Data Collection

The pre- and post-surveys were conducted before and after the program with 30 preceptor nurses.

To measure the Clinical Core Competency of Preceptors (CCCP), the clinical core competency evaluation tool developed by Kwon and colleagues [30] was used. This tool consisted of 10 questions for role models, 8 questions for socialization facilitators, and 16 questions for educators (evaluation of learning needs (4 questions), learning experience planning (3 questions), learning plan execution (7 questions), and task performance evaluation (2 questions)). Responses were measured using a 4-point Likert scale ranging from 1 (not at all) to 4 (strongly agree), with higher scores indicating higher competence. In the original study, Cronbach’s alpha coefficient was 0.96 [30], and it was 0.96 in the present study.

Based on the Interpersonal Communication Competence (ICC) developed by Rubin [31], the General Interpersonal Communication Competence Scale (GICC-15) was developed by Kyung-ho Huh [32] by adding seven concepts. The GICC-15 gives a total score for supportiveness, immediacy, efficiency, social appropriateness, conversational coherence, goal detection, responsiveness, and noise control. It consists of 15 items. Responses were measured on a 5-point Likert scale ranging from 1 (not at all) to 5 (strongly agree), with a higher score indicating higher communication skills. The Cronbach’s alpha coefficient was 0.72 in Huh [32] and 0.86 in the current study.

Qualitative data were collected through participatory observation notes and a weekly reflection journal written by nurses. Participatory observation was performed in all phases of the PAR, including the two weeks of the development phase, when the standards for nursing practice and programs were established, as well as the implementation phase of six weeks. After participating in the weekly preceptor training program for eight weeks, the nurses were asked to complete a reflection log with structured questions.

#### 2.4.2. Data Analysis

Collected quantitative data were analyzed at a significance level of *p* < 0.05 using SPSS/WIN 25.0 (SPSS Inc., Chicago, IL, USA), and the normal distribution of the main variables was confirmed using the Kolmogorov–Smirnov test. Frequencies, percentages, means, and standard deviations were used to describe participants’ general characteristics. A repeated measures analysis of variance with within-subjects factors were performed to assess the effect of the program. Comparisons between pre- and post-study scores regarding CCCP and GICC-15 were performed using t-tests. The Cronbach’s α coefficient was used to validate the measurement reliability.

Regarding qualitative data, the participant observation notes were analyzed by dividing them into planning, execution, evaluation, and reflection to express the research results in a way that others could understand; this was used as a supplement in writing this manuscript. The contents of the reflection journal were categorized through the Krippendorff content analysis procedure [33]. The final category was derived by continuously comparing phrases and sentences belonging to similar categories. In addition, to check the characteristic changes in the educational content and experience, a frequency analysis of the keywords was performed on the weekly reflection journal written by the preceptor nurses. The frequency patterns of the keywords shown in the reflective journal were visualized using WordClouds.

### 2.5. Rigor

The four criteria suggested by Lincoln and Guba [34] were used to ensure the rigor of the research results. First, to secure reliability, we not only actively interacted with education specialist nurses and preceptor nurses during the research process but also observed and recorded their experiences. In the process of exploring the experiences of the development and application of the PEP through the practice research approach, we tried to accept the experiences of the participants as they were by bracketing stereotypes and prejudices. The results for categorizing and analyzing the reflection journal written by the preceptor nurses were shown to the participants, and we checked whether the meaning was accurately conveyed. Second, to ensure suitability, participant observation notes and analytical notes prepared by the researchers were read and analyzed several times. Other educational nurses and preceptor nurses who did not participate in the study were also asked if the experiences of the study participants were consistent with theirs. Third, to ensure auditability, the data collection and analysis process was described in detail and analyzed after confirmation with fellow researchers. Fourth, to ensure corroboration, we tried to draw out the research results in a neutral way by excluding the prejudices of researchers. In particular, in order to maintain neutrality in the analysis process of the reflection diary, the researchers attempted to exclude personal bias by maintaining a certain distance while forming a rapport with the participants. In addition, during the analysis, the writer of the reflection log was treated as anonymous, so that it was not possible to personally check which nurse had written it. The analysis was done by reflecting the participants’ experience shown in the text as much as possible, and the researchers’ past experiences were not considered in the study. In the content analysis process, when there was a disagreement when naming a category, four researchers agreed and reviewed the theme to select the most suitable name to convey the meaning.

### 2.6. Ethical Consideration

Prior to data collection and research, this study received approval (no. CNUH-2020-056) from the IRB of the first three authors’ institution. With the cooperation of the Ministry of Nursing, the purpose of the study, research contents, and procedures were explained to the education specialist nurses and preceptor nurses. The requirements and expectations of the action research were clearly shared. Participants who had volunteered to be part of the study were recruited.

## 3. Results

### 3.1. General Characteristics of the Participants

The general characteristics of the preceptor nurses who participated in this study are shown in Table 1. Twenty-nine participants were women (96.7%) and 17 participants (56.7%) had previous preceptor experience. All participants voluntarily participated in this study, but none of the study participants voluntarily assumed the role of preceptor.

### 3.2. The OMP-Based PEP’s Effect

The results of the changes in the CCCP and GICC-15 for preceptor nurses, after the PEP based on the OMP model had been applied, are shown in Table 2. The CCCP increased from 3.74 (±0.42) to 3.76 (±0.32), and among the subdomains, the facilitator and educator scores also increased after application of the program, but this was not statistically significant. The GICC-15 decreased from 51.67 (±4.77) points to 51.41 (±4.73) points, but this also was not statistically significant.

### 3.3. The Experience of Participants throughout Content Analysis of the Reflective Journals

The results of the content analysis concerning the experiences of participating in the development and operation of educational programs of preceptor nurses are shown in Table 3. Participation experiences were grouped into six key categories: (1) designated as a preceptor without preparation; (2) commitment to do well; (3) camaraderie; (4) growth; (5) awareness of limitation; (6) exhaustion.

The first core category, “Designated as a preceptor without preparation”, had the following subcategories: “Designated preceptor owing to the sudden resignation of a fellow nurse”, “Designated preceptor owing to turnover”, and “Designated preceptor owing to the suggestion of the unit manager”. Although preceptor nurses were randomly assigned without preparation, they voluntarily participated in the participatory practice study and experienced OMP-based PEP. Through the participation experience of OMP-based PEP, they were determined to do well and experienced self-learning and exploration while reviewing and reflecting on their experience in the study with a facilitator.

This specific experience was associated with the second core category. “Commitment to do well” had two subcategories, “Commitment within the PAR team” and “Pledge with the preceptor”. Some of the participants were first-time preceptors, while others had preceptor experience. However, this was the first time that preceptors had experienced facilitators on a PAR team that supported and encouraged them. The experience of pledging their commitment in front of their facilitators and colleague preceptors both presented them with a foreign experience and caused them to rely on the memory of this experience in difficult situations.

The third core category, “Camaraderie”, had two subcategories, “Strong support team” and “Formation of consensus among preceptors”. Preceptors encountered camaraderie while experiencing OMP-based PEP operated by the PAR team for a total of 8 weeks. The preceptors met weekly and felt that they were on a difficult journey together as they looked back and reflected on the activities of the facilitators and preceptors. Moreover, listening to other preceptors’ stories, they felt that it was a problem that everyone was experiencing and they bonded while thinking about solving the problem together.

The fourth core category, “Growth,” involved “Opportunities for learning” and “Check of direction” as subcategories. Learning appeared in the process that participants experienced when solving problems in the PAR process. Preceptors actually taught the OMP concepts that they had learned together through the facilitator. Learning took place during the process of reviewing the literature and finding evidence while developing a practical standard, as well as in the effort to guide new nurses based on OMP. When teaching new nurses by applying OMP-based PEP, there were some doubts as to whether the preceptors were teaching themselves properly. However, during the course of meeting and discussing experiences with the facilitators and fellow preceptors each week, the researchers were able to feel more confidence in the process.

However, there were not only positive aspects to their participation experience. “Awareness of limitations”, which appeared as the fifth core category, involved “Lack of training”, “Difficulties providing positive feedback”, and “Lack of critical thinking” as subcategories. It was not easy to apply the concept of OMP learned from PEP. One of the most difficult things for them was positive feedback. Among the concepts of OMP, providing positive feedback is something that can be done with care and affection. Moreover, preceptors teach new nurses critical thinking. The researchers felt that thinking required conscious practice and effort.

The final core category was “Exhaustion”, involving the subcategories “Repeated feedback for new nurses”, “Interference by senior nurses”, and “Self-blame”. The preceptors had weekly discussions with the PAR team and had positive experiences with the facilitators, but they grew tired of teaching new nurses for 6 weeks. Although the new nurses were taught by applying the OMP concept, the repeated mistakes and questions of the new nurses exhausted the preceptors. There were also cases in which nurses older than themselves ignored the OMP concept and assigned or interfered with new nurses in a personal way. Most of all, the preceptors struggled with the idea that it was their fault when the new nurse did not fully perform their duties.

### 3.4. Feelings and Thoughts about Preceptorship throughout Frequency Analysis of Keywords about Reflective Journals

WordCloud images visualized the frequency analysis results for the keywords from reflective journals, as shown in Figure 2. A total of 541 keywords were extracted from 107 (59.4%) responses regarding preceptors’ feelings or thoughts about their preceptorship. Among them, “recurring education” was most frequently expressed (n = 31, 5.7%), followed by “busy work” (n = 25, 4.6%), “patient” (n = 24, 4.4%), and “practical training” (n = 22, 4.1%). Some examples of participants’ quotations from reflective journals are as follows: “*It is not easy to train new nurses repeatedly”; “Due to the busy work of the ward, continuous education was difficult”; “It is difficult to train a preceptor while caring for a patient”; “I can’t teach it perfectly, so I have to concentrate on the practical training”.*

## 4. Discussion

### 4.1. PAR Approach for Development and Application of OMP-Based PEP

Many studies have addressed the need for changing the existing healthcare practices in Korea [5,8,9,11,12,13]. However, they have mostly focused on improving pain management, geriatric health, and the development of nursing protocols for patients with delirium [35]. The current PAR research is novel since it employs an integrated research method to check the implementation process and effects of developing and applying an OMP-based PEP to foster preceptor competency. This study attempted to objectively evaluate the effectiveness of quantitative and qualitative data through comparative verification and integrated analyses.

As this study was the first PAR conducted at tertiary general hospitals in this region, it was difficult to draw a consensus on the research purpose and research method in the PAR planning stage. Since the nurse educators were accustomed to the top-down educational contents approach, even within the PAR team, it was not easy to reach an agreement because of differing viewpoints. However, participants came to understand each other’s differences, and the active support of the Ministry of Nursing prevented this study from becoming a one-time, fragmentary educational program. To initiate substantial changes and strengthen preceptor capabilities, PAR was used to develop a practical program. This practical research approach resulted in both personal and organizational development [21,22,23,24].

### 4.2. Evaluation of OMP-Based PEP throughout Quantitative and Qualitative Analysis

As action studies have limitations in evaluating effects due to using only a quantitative approach, a mixed study design is often used [28,36]. Our quantitative analysis yielded no significant differences in CCCP or GICC-15. Looking at previous studies, there is a lot of education for new nurses, but it is difficult to find education for preceptors who teach new nurses. Therefore, there were many difficulties in comparing the study results, and only the CCCP and GICC-15 results of the preceptors participating in this study could be compared. In previous research [25], the average CCCP score was 3.94, which is higher than those of our participants because they all had experience as preceptors. In the previous study [37], the average score of GICC-15 of preceptors was 3.34, which was lower than those of our participants. In another study [38], the average score of GICC-15 of preceptors was 3.71, which was higher than those of our participants. In the future, it will be necessary to also check whether core competencies and communication skills improve as the preceptor’s experience increases. In addition, development of the preceptor education program that enhances communication skills is necessary as a strategy to improve the core competency of preceptors.

Although the CCCP tool used in this study was developed for preceptor nurses, looking at the subitems, there is no evaluation item for reflective ability [30]. In previous studies on preceptorship programs, there was no consideration of the preceptors’ reflective abilities [5,7,9,38,39]. However, based on the results of the content analysis in this study, it was confirmed that preceptor nurses grew through reflection during the process of developing and applying OMP-based education programs. Therefore, it is necessary to reexamine reflection in the process of developing and evaluating educational programs to encourage nurses’ learning and to nurture clinical competencies [14].

In addition, the fact that there was no change in GICC-15 regarding the communication ability in this study shows the limitations of the quantitative approach and the self-report questionnaire. Looking at the results of the content analysis, it was confirmed that self-reflection was emphasized throughout the PAR process and that participants recognized their own shortcomings. It is believed that they responded more negatively to the questionnaire. They recognized their lack of communication skills, which they had not previously acknowledged while participating in other educational programs. The results of the content analysis reveal that whenever new nurses made a mistake, the participants thought that it was because they could not properly communicate the educational contents. In a previous study, the preceptor nurses recognized that new nurses had effective and positive communication when they received training and applied and performed well in the clinical field [5,40].

### 4.3. Participation Experience of OMP-Based PEP

The participation experience shown in this study could be confirmed from the results of the content analysis. Although they were designated as preceptors without preparation, preceptors had positive experiences with active support and help from the facilitators of the PAR team. The specific details are outlined below.

Participation in the research was voluntary, although ironically none of the participating preceptor nurses had a choice when they were designated as preceptors within their departments. It is, therefore, inevitable that most preceptor nurses are unprepared and under-skilled for this role. It is not uncommon to assume the role of a preceptor in this unprepared state [5,11,12]. Nurses who do not voluntarily support the role of a preceptor feel burdened with their work and are placed in a stressful situation [11]. It is necessary to help preceptors prepare to teach new nurses through systematic preceptor education and competency-building programs [14].

Participants make a commitment as the first step when participating in OMP-based PEP. Commitment to facilitators and fellow preceptors in PAR was of great significance to them. This commitment was not experienced in the existing preceptor education program. Although simple, it was also an essential step to fulfill the role of the next preceptor. Other studies have shown similar results [21]. When developing and applying other educational programs in the future, having the participants make a commitment before receiving training could help them to perform their assigned tasks more responsibly [18,19,20,21].

Camaraderie was established with facilitators, and colleague preceptors met through the PAR approach. Unlike in the past, when the preceptors were lonely, they had a strong support team and a consensus was formed among preceptors. This feeling of empowerment helps preceptors in overcoming difficulties [14]. Active support at the organizational level and the creation of a positive hospital culture in the organization are necessary for preceptors to fulfill their roles well [13,14].

In the first stage of the OMP model, they were committed to the implementation research process. The second stage comprised composing the work standard draft and training program. In the third stage (i.e., six weeks of practical application), they learned each other’s general rules, including accurate work standards and evidence-based nursing skills. In stages four and five, feedback was provided to each other regarding the strengths and shortcomings at each conference. In this process, they grew one step further by checking the direction of preceptorship and trying to compensate for the shortcomings [18,19,20,21]. In nursing practice, closely observing and giving feedback to new nurses was also a process of growth through making efforts to acquire positive feedback and constructive feedback skills for preceptors [17,18,19].

Preceptors also had negative experiences with OMP-based PEP, which were “awareness of limitations” and “exhaustions”. One of the things emphasized in PAR is reflection, and in the process of reviewing their activities every week and discussing them with the facilitator, they experienced a lot of limitations. The researchers learned the OMP concept, although it took a conscious effort to put it into practice. Further, it was difficult for preceptors because they were not used to giving positive feedback to new nurses. Preceptors know the importance of critical thinking, although it has been difficult to communicate it logically. Previous studies have also shown that the core competency of preceptors is affected by critical thinking ability [37]. Therefore, in the future, specific training to increase critical thinking ability should be considered as a preceptor education strategy [37].

In addition, an interventional approach is required, considering the cases where preceptors experience exhaustion [12]. The key to the OMP concept is to deliver the key content in a short time and to teach through feedback [18,20], although this was hard to do. When a new nurse could not do their job completely, the preceptor’s guilt and self-blame was exacerbated, causing exhaustion. In order for the preceptor to actively apply the OMP concept in practice, it is necessary not only to have an empowering atmosphere, but also to help the preceptor nurse avoid feeling that the preceptee’s mistakes are the responsibility of the preceptor [7,9,10].

The frequency analysis results of the keywords from reflective journals were similar those of content analysis, although with different contents. Since the words most frequently used in the reflection journal were extracted, they mainly represented the concepts that preceptor nurses thought the most. The feelings and thoughts that appeared while participating in PEP were recurring education, a concept that appeared in content analysis. “Busy work” and “patient” were not mentioned separately in the content analysis, but they are concepts that participants consider after work every day. These words came from the thought that it was not easy to act as a preceptor due to busy work, and it was difficult to act as a preceptor while focusing on patients. Based on this analysis, in order to expand and operate OMP-based PEP, preceptor nurses will have to reduce the number of patients compared to other nurses or organize and operate them with personnel other than the working manpower [11,12]. In addition, although the preceptor nurse knows that practical training should be given to new nurses, it is necessary to operate the PEP while considering that practically caring for patients involves many limitations [9,12].

### 4.4. Implications for Practice

In Korea, the 2019 New Nurse Guidelines recommends three months for new nurse training programs [41]; however, this is difficult in reality because it requires a large budget and considerable time to operate the preceptorship for three months. This study’s eight-week preceptorship was more effective and efficient.

The action research process provided opportunities to examine the problems faced by the participants and researchers in clinical practice. Discussing these problems with other practitioners not only provided strategies for effective implementation of preceptorship programs in the future but also provided basic data for developing field-oriented educational programs. According to a previous study [42], 55.0% of new nurses were confused by the education contents. In addition, the participatory practice research approach to PEP development differs from traditional education programs. Rather, the program promotes the development of cooperative relationships between nurse educators and preceptor nurses. The importance of action research lies in the fact that researchers and clinical practitioners work cooperatively with each other, such as sharing thoughts and solutions in specific areas and empowering each other [21,22,43,44].

It is also meaningful that this study provides an opportunity to explore effective ways to utilize education specialist nurses provided through support projects that are being piloted in Korea [41]. The implementation of the education-only nurse system aims to alleviate the workload of practicing nurses and reduce the turnover rate of new nurses—who must go through a series of processes, such as training opportunities, operation, and evaluation [15,16]. Since this system is relatively new, it is thought that a more successful system can be established only after various trials and evaluations. Therefore, this study is significant in that it expands the role of nurses in charge of education, highlights the need for continuing education among preceptor nurses, and promotes reflection by new nurses concerning their roles. Studies on the OMP program [39] have often used a single research method to evaluate its effectiveness and often found it ineffective in clinical practice. However, the mixed research method used in the present study has been able to verify its advantages [20,21]. In the future, the results of this study can be used as basic data for developing programs for nurses to actively change their practice and solve their problems through a participatory, action-research-based approach to clinical practice.

### 4.5. Limitations

This study had some limitations. First, because it was conducted with nurses at tertiary general hospitals in one region, the results can only be interpreted within the cultural context of Korea and the generalizability is limited. Second, this study did not include a control group and it was difficult to verify the effects of PEP.

## 5. Conclusions

The PAR revealed that the OMP-based PEP is an effective strategy for capacity building among nurses. Further, such strategies are required to support nursing educators and nursing departments. Systematic and supportive programs for improving nurse competency is important for the preceptorship. Similarly, support and encouragement for preceptor nurses are important for effective preceptorship operations, and it is necessary to help them at the organizational level. In addition, PAR can be used as a very effective intervention strategy to help clinical nurses learn and grow.

This study was conducted at a tertiary general hospital in Korea to identify solutions to resolve their nurse turnover issues. Given the serious shortage of nurses worldwide, the OMP-based preceptor competency training program developed in this study provides a strategy that can be replicated by various healthcare organizations to improve nursing effectiveness. The research data can also be used to devise a basic utilization plan. In addition, the PAR process promotes learning and growth among nursing educators and preceptor nurses, even when they were burdened by a heavy workload and busy conditions. This program can promote competency among not only new nurses but also experienced nurses, an aspect that has widespread implications for the clinical nursing field. Therefore, it is necessary to encourage OMP activities for preceptors through an action research approach and to support research and practice in clinical settings. Organizational support can reduce their burden.

## Figures and Tables

**Figure 1 ijerph-18-11376-f001:**
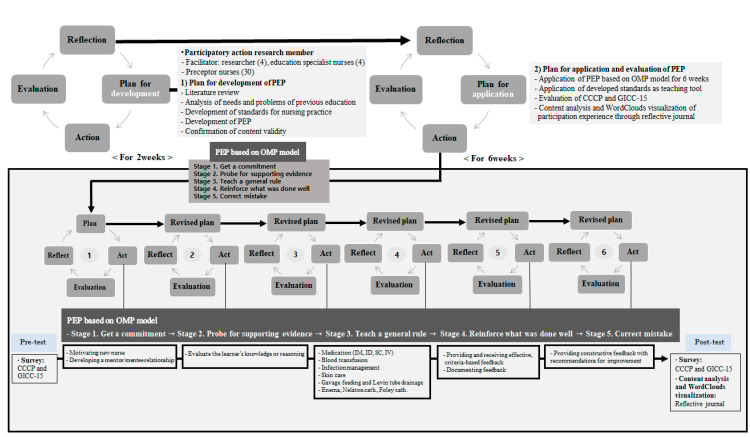
PEP based on the OMP model through PAR.

**Figure 2 ijerph-18-11376-f002:**
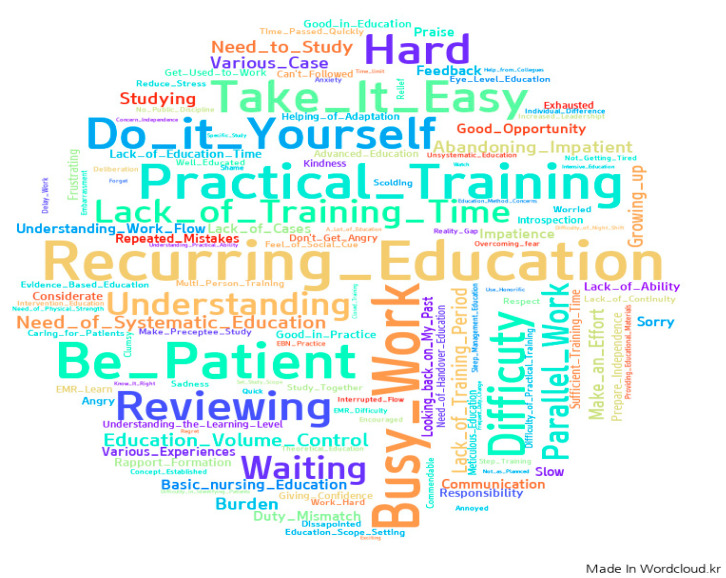
Feelings and thoughts regarding preceptorship, as shown in the reflective journals.

**Table 1 ijerph-18-11376-t001:** Participants’ General Characteristics.

Characteristics	Categories	N (%)	Mean (SD)
Age (years)	<30	20 (66.7)	28.8 ± 2.30
	≥30	10 (33.3)	
Sex	Men	1 (3.3)	
	Women	29 (96.7)	
Marital status	Single	26 (86.7)	
	Married	4 (13.3)	
Types of religion	Protestantism	5 (16.7)	
	Catholicism	3 (10.0)	
	None	21 (70.0)	
	Others	1 (3.3)	
Total working career (years)	≤3	10 (33.3)	
	4–5	11 (36.7)	
	≥6	9 (30.0)	
Present unit career (years)	<3	6 (20.0)	
	3–4	12 (40.0)	
	≥5	12 (40.0)	
Work unit	Ward	13 (43.3)	
	Intensive care unit	17 (56.7)	
Preceptor experience	Yes	17 (56.7)	
	No	13 (43.3)	
Reason of preceptor	Recommendation from unit manager	30 (100.0)	
	Voluntary	0 (0)	

(N = 30).

**Table 2 ijerph-18-11376-t002:** Effects of PEP on the variables.

Variables	Pre-Test Mean (SD)	Post-Test Mean (SD)	*t*	*p*
Clinical Core Competency of Preceptors (CCCP)	3.74 ± 0.42	3.76 ± 0.32	0.29	0.777
Role model	3.91 ± 0.43	3.89 ± 0.34	0.35	0.733
Facilitator	3.65 ± 0.50	3.69 ± 0.42	0.45	0.655
Educator	3.69 ± 0.40	3.72 ± 0.35	0.42	0.675
Learning needs assessment	3.76 ± 0.50	3.78 ± 0.35	0.26	0.794
Learning experience planning	3.52 ± 0.56	3.53 ± 0.53	0.06	0.954
Learning plan implementation	3.75 ± 0.42	3.83 ± 0.39	0.84	0.410
Job performance evaluation	3.55 ± 0.62	3.50 ± 0.54	0.45	0.655
General Interpersonal Communication Competence (GICC-15)	3.46 ± 0.45	3.43 ± 0.30	0.44	0.666

(N = 30).

**Table 3 ijerph-18-11376-t003:** Content analysis of participation experiences from reflective journals.

Category	Subcategory	Quotation
Designated as a preceptor without preparation	Designated preceptor owing to the sudden resignation of a fellow nurse	Rather than participating voluntarily; I got the preceptor job really suddenly. As a fellow nurse who was supposed to be a preceptor resigned, I was designated owing to similarity in age with the previous preceptor.—Participant 3
	Designated preceptor owing to turnover	Since it was my turn, I had no choice but to become a preceptor.—Participant 8
	Designated preceptor owing to same work duties as new nurses	The preceptor had the rotation of departments, and I had to fill the vacant position. That day, I was assigned because I had the same work schedule as a new nurse.—Participant 2
	Designated preceptor owing to the suggestion of the unit manager	The head nurse told me to do it by designation, but I could not refuse. A preceptor is not something you can do selectively.—Participant 6
Commitment to do well	Commitment within the PAR team	The pledge within the practice research team to help for my preceptorship made me resolve to work even harder.—Participant 4
	Pledge with the preceptor	Not a promise I made alone, but a promise I made with someone made me teach more responsibly.—Participant 10
Camaraderie	Strong support team	When we got together concurrently every week and had time to look back on our preceptor activities for the week, I thought we were going together, not alone.—Participant 23
	Formation of consensus among preceptors	Listening to the stories of nurses doing preceptor with me, it was comforting to know that it was not just my problem, but that everyone faces the same difficulties.—Participant 8
Growth	Opportunities for learning	I was able to have time to look back on the week by writing a reflection journal every week, and I think I learn as I teach.—Participant 1
	Check of direction	While checking whether I was teaching properly, I checked the direction for correct preceptor education. And when I found something was lacking, I made an effort to make up for it.—Participant 16
Awareness of limitation	Lack of training	I knew I had to provide short, positive, and constructive feedback with the essentials, but I regretted not doing it because I wasn’t trained.—Participant 14
	Difficulties providing positive feedback	I was taught to provide positive feedback to new nurses, but that didn’t work out well in an urgent clinical situation. It was really hard to praise.—Participant 20
	Lack of critical thinking	I had to teach new nurses to think critically about why this happens, but it didn’t work out. I think we have to practice deliberately and consciously.—Participant 18
Exhaustion	Repeated feedback for new nurses	I was tired of seeing a new nurse who learned from me and who I gave feedback repeatedly to several times but who still couldn’t do it.—Participant 15
	Interference by senior nurses	Even if I try to teach systematically according to the standards, if a higher-year nurse gives a different way to a new nurse, I have to follow it.—Participant 12
	Self-blame	It was not easy to teach as I was taught from the PAR team, so I didn’t teach well. So, it seems that new nurses can’t learn to work.—Participant 25

## Data Availability

The data presented in this study are not publicly available because of privacy concerns.

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
