# Peer review of "Development and Evaluation of a Preceptor Education Program Based on the One-Minute Preceptor Model: Participatory Action Research"

_ijerph, 2021, doi:10.3390/ijerph182111376_

Round 1
Reviewer 1 Report
My general opinion of the article:
I think the clinical purpose is too ornate to be a simple skill acquisition activity. I understand that the pandemic has made it very difficult to do so.
Based mainly on the conclusions and results, I think that their conclusions are excessively resounding. It is an activity for the adaptation of new nurses, which due to the pandemic, the rotation and the shortage has increased. Organizations should promote these activities, which surely there are many others.
I think they could have increased the sample. The qualitative part seems perfect to me, but the conclusions I think should be more modest / ambiguous. It could also be raised with auxiliary personnel, for example.
The conclusion that they give in the ABSTRACT I would ask them to modify it.
Author Response
Title: “Development and Evaluation of a Preceptor Education Program Based on the One-Minute Preceptor Model: Participatory Action Research” (ijerph-1375587)
We recently submitted an original research article entitled “Development and Evaluation of a Preceptor Education Program Based on the One-Minute Preceptor Model: Participatory Action Research” for publication in the International Journal of Environmental Research and Public Health (ijerph-1375587). We are grateful to you and the journal’s reviewers for the comments on the original version of our manuscript. Please see our responses to the reviewers’ comments submitted with this letter.
We revised the manuscript based on the review reports. All changes were highlighted, and responses to reviewers’ comments are included with this cover letter. We hope that the changes we made have significantly improved the quality of our manuscript. We hope that the explanations and revisions of our work are satisfactory to the other reviewers.
We believe that this is a significant contribution to the existing literature because our results will be useful as basic data in the development of programs and interventions to promote the competencies of preceptor nurses in the severe nursing shortage situation, as well as in the development of strategies to apply the participatory action research approach to clinical practice management.
We look forward to hearing from you!
Best Regards,
Suhyun Kim
Department of Nursing, Nambu University
23 Cheomdanjoongang-ro, Gwangsan-gu
Gwangju 62271, Republic of Korea
Tel.:82-62-970-0249; Fax:82-62-970-0261
E-mail: ksh136112@gmail.com

Reviewer 2 Report
Review of ijerph-1375587
Development and Evaluation of a Preceptor Education Program based on the One-minute Preceptor model: Participatory Action Research
This manuscript presents the results of a Participatory Action Research (PAR) study conducted in a tertiary hospital in Korea.
Does the introduction provide sufficient background and include all relevant references?
The Introduction and Literature Review are one under the subheading of Introduction. Possibly the authors wanted to merge them and to include all the elements of both in the same section. However, the Literature Review section is extremely short and weak. The authors talk about “Another resource is the One-minute Preceptor (OMP), a preceptor training model, developed to train medical residents and students” (line 56). A more complete explanation of how the model was developed and why the researchers decided to use it for a population for which was not developed is warranted. Then, line 68, states “The OMP model is used to improve clinical reasoning and preceptor feedback skills in the clinical field. It is preferred over traditional education methods.” What are these “traditional education methods”? In a few words, the Review of Literature needs to be expanded to include more explanation on the reasons why the study was conceived, and on the decisions to use the specific methods and instruments selected for this study.
Is the research design appropriate?
The research design seems to be appropriate, but it lacks detail. The authors mention first and second research teams (lines 128-130) but it is unclear what their functions were and if both teams participated in the entire study. The manuscript includes a table (Table 1) with the participants’ demographic information. Again, there is no explanation why the data was collected, if it was used, and how it was used. The researchers explain a research design that includes four stakeholders: researchers, education specialists, preceptors, and new nurses. However, little to no data is reported on the researchers and the new nurses.
Are the methods adequately described?
The methods are not clearly described. For example, the paragraph between lines 196 and 205 is difficult to understand because the description of Rubin’s interpersonal communication competence and the Comprehensive Communication Ability Scale are mixed. It is difficult to understand which instrument has which competence. The Data Analysis also lacks explanation. The authors included a section on rigor with all the right words but vague in content. An example is found on line 246 “data collection and analysis process was described in detail and analyzed after confirmation with fellow researchers.” Unfortunately, such detail was not included in the manuscript. Then, line 248, it states “we tried to draw out the research results in a neutral way by excluding the prejudices of researchers.” Again, none of this is explained. Also missing is a more detailed process of how the researchers were able to exclude their prejudices, which any qualitative researcher knows is virtually impossible to do so.
Are the results clearly presented?
The Results are not clearly presented. As well as the rigor for the study was presented, data collected to achieve such rigor did not seem to find their way into the discussion of the results. Table 2, Effects of PEP on Variables, was mentioned almost in passing and it was not discussed. Themes and quotes were presented but there was not a detailed explanation as to how the researchers arrived at such themes. The Discussion attempts to discuss the qualitative data results but not in a systematic way, at least not category by category. It is also not clear what data was used to generate the word visualization. At the beginning of the Results section, it was mentioned “WordCloud images visualized the content analysis results, as shown in Figure 2. A total of 541 keywords was extracted from 107 (59.4%) responses about preceptors’ feelings or thoughts about their preceptorship (lines 262-263). Were these responses taken from the qualitative scales? The description that accompanies Figure 2 reads “Figure 2. Feelings and thoughts about preceptorship, as shown in the reflective journals.” Where did the data come from?
In the Discussion section, the researchers moved from quantitative to qualitative results trying to find which one better fits their results. However, the research questions are never restated, and they are not clearly answered. It will be good to restate the questions and to offer a response. The researchers use the words evidence-based practice in the Discussion section, but the words were not previously identified during the statement of the problem, data collection, or data analysis. Perhaps if the researchers take a more systematic approach to discuss the results, the manuscript would be easier to understand.
Are the conclusions supported by the results?
The Conclusions section is very short and not descriptive enough. It would seem that the researchers used the Discussion section to discuss the Conclusions. The Conclusion section needs to be expanded and it needs to discuss “lessons learned” and how those lessons can be applied for future research.
Author Response

(The authors gave the same response as above.)

Round 2
Reviewer 2 Report
Dear Researchers:
Thank you for addressing all my concerns. There is only one thing left. In the discussion of the themes, you added the participant number, which is great. However, there is a conflict on the participant number on the following:
Repeated feedback for new nurses |
I was tired of seeing a new nurse who learned from me and who I gave feedback repeatedly to several times but who still couldn't do it. - Participant 15’s reflective journal. - Participant 26 |
As you see, you had mentioned participant 15 but then you identified it as participant 26. Other than that, I believe the paper is worthy of publication.